# Obesity and Dyslipidemia Synergistically Exacerbate Psoriatic Skin Inflammation

**DOI:** 10.3390/ijms23084312

**Published:** 2022-04-13

**Authors:** Kenta Ikeda, Shin Morizane, Takahiko Akagi, Sumie Hiramatsu-Asano, Kota Tachibana, Ayano Yahagi, Masanori Iseki, Hideaki Kaneto, Jun Wada, Katsuhiko Ishihara, Yoshitaka Morita, Tomoyuki Mukai

**Affiliations:** 1Department of Dermatology, Okayama University Graduate School of Medicine, Dentistry, and Pharmaceutical Sciences, Okayama 700-8558, Japan; pbhb4xbu@s.okayama-u.ac.jp (K.I.); zanemori@cc.okayama-u.ac.jp (S.M.); py4r2ift@okayama-u.ac.jp (K.T.); 2Department of Immunology and Molecular Genetics, Kawasaki Medical School, Kurashiki 701-0192, Japan; yahagi@med.kawasaki-m.ac.jp (A.Y.); miseki@med.kawasaki-m.ac.jp (M.I.); ishihara-im@med.kawasaki-m.ac.jp (K.I.); 3Department of Rheumatology, Kawasaki Medical School, Kurashiki 701-0192, Japan; akagitahiko@gmail.com (T.A.); h061eb@gmail.com (S.H.-A.); morita@med.kawasaki-m.ac.jp (Y.M.); 4Department of Diabetes, Endocrinology and Metabolism, Kawasaki Medical School, Kurashiki 701-0192, Japan; kaneto@med.kawasaki-m.ac.jp; 5Department of Nephrology, Rheumatology, Endocrinology, and Metabolism, Okayama University Graduate School of Medicine, Dentistry, and Pharmaceutical Sciences, Okayama 700-8558, Japan; junwada@okayama-u.ac.jp

**Keywords:** psoriasis, obesity, dyslipidemia, leptin, palmitic acid, chemokine

## Abstract

Patients with psoriasis are frequently complicated with metabolic syndrome; however, it is not fully understood how obesity and dyslipidemia contribute to the pathogenesis of psoriasis. To investigate the mechanisms by which obesity and dyslipidemia exacerbate psoriasis using murine models and neonatal human epidermal keratinocytes (NHEKs), we used wild-type and *Apoe*-deficient dyslipidemic mice, and administered a high-fat diet for 10 weeks to induce obesity. Imiquimod was applied to the ear for 5 days to induce psoriatic dermatitis. To examine the innate immune responses of NHEKs, we cultured and stimulated NHEKs using IL-17A, TNF-α, palmitic acid, and leptin. We found that obesity and dyslipidemia synergistically aggravated psoriatic dermatitis associated with increased gene expression of pro-inflammatory cytokines and chemokines. Treatment of NHEKs with palmitic acid and leptin amplified pro-inflammatory responses in combination with TNF-α and IL-17A. Additionally, pretreatment with palmitic acid and leptin enhanced IL-17A-mediated c-Jun N-terminal kinase phosphorylation. These results revealed that obesity and dyslipidemia synergistically exacerbate psoriatic skin inflammation, and that metabolic-disorder-associated inflammatory factors, palmitic acid, and leptin augment the activation of epidermal keratinocytes. Our results emphasize that management of concomitant metabolic disorders is essential for preventing disease exacerbation in patients with psoriasis.

## 1. Introduction

Psoriasis is a chronic inflammatory skin disease characterized by erythematous plaques with silvery scales. The pathogenesis of psoriasis predominantly involves epidermal hyperkeratosis associated with dysregulation of interleukin (IL)-17A-mediated immune responses in inflamed skin lesions [1]. Clinical and epidemiological studies have demonstrated a close association between psoriasis and various metabolic disorders, such as obesity, metabolic syndrome, dyslipidemia, diabetes mellitus, and hypertension [2]. A meta-analysis of 16 observational studies showed the pooled odds ratio for the association between psoriasis and obesity to be 1.66 (95% confidence interval: 1.46–1.89) [3]. In a systematic review, 20 of 25 included studies revealed significant associations between psoriasis and dyslipidemia, with odds ratios ranging from 1.04 to 5.55 [4]. These data suggest the involvement of metabolic disorders in the pathogenesis of psoriasis.

Obesity is a chronic inflammatory condition accompanied by metabolic dysregulation [5]. Several obesity-related factors, including adipokines and fatty acids, regulate inflammatory responses [6,7,8]. For example, adipocytes become enlarged in obesity and, subsequently, produce large numbers of adipocytokines, which create an inflammatory environment in the body and lead to the exacerbation of systemic chronic inflammatory diseases [9]. In this context, the potential relationship between obesity and psoriasis has gradually been elucidated [10]. The severity of psoriasis has been associated with the severity of obesity [3]. Serum leptin levels are increased in psoriasis patients [11]. Studies of the murine psoriasis model have revealed that obesity augments psoriatic skin changes, and that adipokines and palmitic acid are pathologically involved [12,13,14].

Independent of obesity, dyslipidemia is also clinically associated with psoriasis [15,16]. The pathological impact of dyslipidemia on psoriasis has been investigated using *Apoe*-deficient (*Apoe*^−/−^ mice, which exhibit severe dyslipidemia without weight gain. Dyslipidemia exacerbated hyperkeratosis in an imiquimod (IMQ)-induced psoriasis model [17]. Considering the fact that *Apoe*^−/−^ mice do not exhibit obesity, dyslipidemia itself may be an aggravating factor for psoriatic skin changes, independent of obesity. This supposition is supported by the results of multivariate analyses of psoriasis patients, showing that dyslipidemia is a risk factor for psoriasis even after adjusting for body mass index [15].

Patients with psoriasis often have multiple comorbidities [2]. Additionally, the number of metabolic factors has been shown to be positively correlated with the risk of psoriasis [18]. Although individual metabolic conditions can potentially exacerbate psoriatic skin changes in murine models, the synergistic effects of metabolic conditions have not been investigated. In this study, we investigated whether and how metabolic disorders affect the pathogenesis of concomitant psoriatic skin inflammation, using a murine psoriasis model complicated by obesity and dyslipidemia. Furthermore, we investigated the pathogenic effects of metabolic-disease-associated inflammatory factors on the innate immune responses of epidermal keratinocytes.

## 2. Results

### 2.1. High-Fat-Diet-Induced Obesity and Apoe-Deficiency-Induced Dyslipidemia Synergistically Augment Psoriatic Skin Inflammation

To investigate the effects of obesity and dyslipidemia on the development of psoriatic skin inflammation, we used obese mice that were fed a high-fat diet (HFD), *Apoe*-deficient dyslipidemic mice, and dyslipidemic obese mice (HFD-fed *Apoe*^−/−^ mice). Psoriatic skin inflammation was induced by applying IMQ topically on the ears of the mice (Figure 1a). As shown in Figure 1b, HFD-fed mice exhibited obesity, represented by a 30–40% increase in body weight compared to that of normal diet (ND)-fed mice (Figure 1b–d). *Apoe* deficiency did not alter body weight compared to the corresponding wild-type (WT) mice (Figure 1b–d). The serum total cholesterol levels were remarkably increased in *Apo*^−/−^ mice compared to those in WT mice (Figure 1e). A significant increase in triglyceride levels was observed only in HFD-fed *Apoe*^−/−^ mice (Figure 1f). Free fatty acid levels tended to increase in *Apoe*^−/−^ mice (Figure 1g). No significant changes were observed in blood glucose levels between the groups (Figure 1h).

The development of psoriatic lesions was examined after 5-day topical treatment with IMQ. In control groups treated with petrolatum (Vaseline), obesity and dyslipidemia did not significantly affect ear thickness (Figure 2a). IMQ treatment induced erythematous, scaly, and thickened lesions. Neither obesity nor dyslipidemia significantly worsened skin thickening compared to ND-fed WT mice, whereas the coexistence of obesity and dyslipidemia markedly exacerbated skin thickness (Figure 2b). Histopathological analyses revealed that IMQ treatment induced hyperkeratosis, acanthosis, and neutrophilic infiltration, which was prominent in HFD-fed *Apoe*^−/−^ mice (Figure 2c). Quantitative analysis showed a significant increase in epidermal thickness in HFD-fed *Apoe*^−/−^ mice (Figure 2d). Exacerbated inflammatory skin changes in the HFD-fed *Apoe*^−/−^ mice were confirmed by the increased expression of IL-23p19, signal transducer and activator of transcription-3 (STAT3), and phospho-STAT3 in the epidermis of inflamed skin (Appendix A), which are known to be activated in psoriatic skin [19,20]. Collectively, these data indicate that the combination of obesity and dyslipidemia triggered more intense psoriatic inflammation than obesity or dyslipidemia alone.

### 2.2. Coexistence of Obesity and Dyslipidemia Enhances Inflammatory Gene Expression in Psoriatic Lesions

To investigate the mechanisms by which obesity combined with dyslipidemia exacerbates psoriatic inflammation, we assessed the mRNA expression of pro-inflammatory cytokines and chemokines in the inflamed lesions.

The expression of pro-inflammatory cytokines—such as *Il1b*, *Il6*, *Il17a*, *Tnf*, *Il23a*, and *Il22*—was increased in the IMQ-treated groups (Figure 3). Among them, *Il23a*, *Il17a*, *Il6*, and *Il22* were significantly elevated in HFD-fed *Apoe*^−/−^ mice compared to ND-fed WT mice (Figure 3). These patterns were consistent with the severity of histological skin changes in the IMQ-treated group. The expression of *Il17c* and *Il19* was also significantly increased in IMQ-treated, HFD-fed *Apoe*^−/−^ mice (Figure 3). These cytokines are produced by inflamed epidermal keratinocytes, which are known to amplify inflammation and epidermal hyperkeratosis [21,22]. Additionally, chemokine gene expression was significantly elevated in IMQ-treated, HFD-fed *Apoe*^−/−^ mice—especially that of *Ccl20*, *Cxcl1*, *Cxcl3*, and *Cxcl5* (Figure 3), which are known to attract IL17A-producing immune cells and neutrophils [23]. Increased expression of the antimicrobial peptide genes *Defb4*, *S100a8*, and *S100a9*, as well as the keratinocyte proliferation marker *Krt16*, was observed in the IMQ-treated groups (Figure 3), which were not augmented by HFD feeding or *Apoe* deficiency. These findings suggest that activated epidermal keratinocytes enhance chemokine production, leading to amplification of psoriatic aberrant immune responses in the skin.

In the Vaseline-treated groups, the following intriguing trends were observed: *Ccl20* expression was predominantly increased in HFD-fed mice regardless of genotype, and *Il19* expression was predominantly increased in *Apoe*^−/−^ mice regardless of their diet, although these differences were not statistically significant (Appendix A).

Putting these results together, we proposed the following hypotheses: (1) metabolic diseases distinctly predispose patients to psoriatic inflammation, depending on their metabolic status, and (2) epidermal keratinocytes initiate or amplify pro-psoriatic inflammation by responding to metabolic-disease-associated inflammatory factors.

### 2.3. Increased Serum Leptin Levels in HFD-Fed Mice

Adipokines are cytokines produced by adipocytes in obese conditions [24]. Because adipokines regulate inflammatory processes [25,26], we examined serum levels of adipokines, including leptin, resistin, IL-6, monocyte chemotactic protein-1 (MCP-1), and tumor necrosis factor-α (TNF-α). We found that leptin levels were significantly increased in HFD-fed groups regardless of *Apoe* deficiency or IMQ treatment (Figure 4a), which is considered to reflect the obese status, as reported [27,28]. Serum levels of resistin were not altered by HFD feeding or *Apoe* deficiency in Vaseline-treated mice, but were slightly decreased by HFD feeding in IMQ-treated mice (Figure 4b). IL-6 and MCP-1 levels were not significantly altered between the groups (Figure 4c,d). Serum levels of TNF-α were below detectable levels in all groups (data not shown).

We examined whether serum IL-17A levels are increased by metabolic conditions. Serum IL-17A levels were found to be comparable across all groups, except for a slight elevation in IMQ-treated, HFD-fed mice (Figure 4e). Combined with the results showing increased Il17a expression in the skin (Figure 3), IL-17A is likely to be elevated locally in the inflamed skin.

### 2.4. Palmitic Acid and Leptin Regulate Inflammatory Responses of Epidermal Keratinocytes in the Presence of Inflammatory Cytokines

We investigated how metabolic-disease-associated inflammatory factors affect the innate immune responses of epidermal keratinocytes. In the pathogenesis of psoriasis, epidermal keratinocytes produce various inflammatory cytokines and chemokines, leading to the recruitment of immune cells, such as IL-17A-producing cells and neutrophils [29]. To examine the responses of keratinocytes, we cultured neonatal human epidermal keratinocytes (NHEKs) and stimulated them with IL-17A, TNF-α, palmitic acid, and leptin.

Palmitic acid is a saturated free fatty acid known for its pro-inflammatory activity [30]. NHEKs were stimulated with palmitic acid in combination with TNF-α and/or IL-17A. We found that the inflammatory cytokine gene expression was significantly enhanced in response to TNF-α or IL-17A. Additionally, substantial synergistic effects of TNF-α and IL-17A were observed (Figure 5a), consistent with our previous study’s findings [31]. Although palmitic acid alone did not significantly alter gene expression, it significantly amplified TNF-α- and IL-17A-mediated responses—especially the expression of *CCL20*, *CXCL1*, *CXCL8*, *IL1B*, and *TNF* (Figure 5a). Furthermore, palmitic acid treatment prominently augmented *IL19* mRNA expression in the presence of TNF-α and IL-17A (Figure 5a).

Leptin is an adipokine that acts in the hypothalamus to suppress appetite and increase energy expenditure [32]. Since leptin has a pro-inflammatory capacity [33,34], and its levels were elevated by HFD feeding, we examined the pro-inflammatory effect of leptin on NHEKs. NHEKs were treated with leptin in the presence of IL-17A. IL-17A treatment increased the expression of chemokine genes, such as *CCL20*, *CXCL1*, and *CXCL8* (Figure 5b). Although leptin alone did not elevate the expression, it significantly amplified the effect of IL-17A (Figure 5b).

Based on these findings, palmitic acid and leptin augment TNF-α- and IL-17A-mediated inflammatory responses in epidermal keratinocytes. Thus, metabolic disorders could exacerbate psoriatic skin inflammation, presumably via increased palmitic acid and leptin levels.

### 2.5. Palmitic Acid and Leptin Activate the Phosphorylation of C-Jun N-Terminal Kinase Synergistically with IL-17A

We next investigated the mechanisms by which palmitic acid and leptin enhance pro-inflammatory responses in keratinocytes. We initially speculated that inflammatory cytokines, palmitic acid, and leptin could affect the expression of one another’s receptors. NHEKs were treated with TNF-α, IL-17A, palmitic acid, and leptin for 24 h, and gene expression levels for receptors were determined: *IL17RA* and *IL17RC* for IL-17A, *TNFRSF1A* for TNF-α, and *LEPR* for leptin. We observed no substantial changes in the gene expression levels after the indicated stimulation compared to that in non-stimulated cells, except for a slight elevation in *LEPR* after palmitic acid stimulation (Figure 6a).

Next, we hypothesized that intracellular signaling crosstalk could exist under inflammatory stimuli. We examined the downstream signaling pathways of IL-17A in combination with palmitic acid and leptin. NHEKs were pretreated with palmitic acid and leptin for 2 h and then stimulated with IL-17A. We found that IL-17A activates the phosphorylation of c-Jun N-terminal kinase (JNK), p38, and NF-κB p65, whereas palmitic acid and leptin modestly augment the IL-17A-mediated phosphorylation of JNK (Figure 6b and Appendix A). TNF-α also induced the phosphorylation of JNK, p38, and NF-κB p65 in NHEKs. However, pretreatment with palmitic acid and leptin did not noticeably augment the TNF-α-mediated phosphorylation of the pathways (data not shown).

## 3. Discussion

We investigated the synergistic effects of obesity and dyslipidemia on the development of IMQ-induced psoriatic skin inflammation. We used *Apoe*-deficient dyslipidemic mice and obese mice that were fed with an HFD. We found that the coexistence of obesity and dyslipidemia significantly augmented acanthosis in association with increased gene expression of inflammatory cytokines and chemokines. We also found that palmitic acid and leptin enhance inflammatory responses in epidermal keratinocytes in the presence of inflammatory cytokines in vitro.

The coexistence of obesity and dyslipidemia prominently enhanced the severity of psoriatic skin inflammation in a synergistic manner. Previous studies have reported that HFD-induced obesity intensifies psoriasiform dermatitis and demonstrates the potential involvement of leptin [35]. Additionally, *Apoe* deficiency exacerbates skin thickness [17], and palmitic acid and oxidized low-density lipoprotein are possibly involved in inflammation [17,36]. The synergistic effects of obesity and dyslipidemia on psoriatic inflammation correspond to the results of a recent clinical study that demonstrated a positive association between the risk of psoriasis and the number of metabolic factors, including increased waist circumference, high triglyceride levels, low high-density lipoprotein cholesterol levels, high blood pressure, and high blood glucose levels [18].

Leptin is an adipocyte-derived cytokine that modulates inflammatory responses in various pathological conditions, including psoriasis [10,37,38]. Serum leptin levels in patients with psoriasis are reported to be elevated [11]. Leptin acts on diverse immune cells, including dendritic cells, neutrophils, natural killer cells, T cells, and B cells [39], and acts on epidermal keratinocytes, causing proliferation and inflammation of the skin [34,40]. Leptin-deficient (*ob/ob*) mice exhibited reduced skin inflammation in an IMQ-induced psoriasis model [35]. In the current study, serum leptin levels were increased in HFD-fed WT and *Apoe*^−/−^ mice; moreover, leptin treatment significantly enhanced *CCL20*, *CXCL1*, and *CXCL8* expression in the presence of IL-17A in NHEKs. These data indicate that leptin aggravates psoriatic skin inflammation, in accordance with previous studies [33,35,40].

Palmitic acid, a saturated fatty acid (SFA), is reportedly involved in inflammatory processes in some pathological conditions, such as cardiovascular diseases, neutrophilic folliculitis, and psoriasis [41,42,43]. SFAs increase under obese conditions and promote inflammation by affecting various immune cells, including macrophages [44,45]. In relation to the pathogenesis of psoriasis, palmitic acid is reported to indirectly stimulate epidermal keratinocytes [14]. The previous study showed that palmitic acid enhanced LPS-mediated inflammatory cytokine production from macrophages; in turn, secreted inflammatory factors induced chemokine production in epidermal keratinocytes [14]. Although we did not test the effects of palmitic acid on macrophages in this study, we found that palmitic acid directly activates epidermal keratinocytes in the presence of TNF-α or IL-17A. These results indicate that palmitic acid could create an inflammatory milieu in the skin by activating both macrophages and keratinocytes.

Palmitic acid activates intracellular signaling pathways, such as mitogen-activated protein kinases, Toll-like receptor signaling, and protein kinase C [46]. In our experiments, palmitic acid significantly enhanced the expression of *CCL20*, *CXCL8*, and *IL1B* in the presence of TNF-α and IL-17A. In terms of the mechanisms, we found that pretreatment with palmitic acid or leptin augmented IL-17A-mediated JNK phosphorylation in NHEKs. JNK phosphorylation is involved in the downstream pathways of all three stimuli. JNK is activated in the downstream signaling of IL-17A receptors (IL-17RA/IL-17RC) and leptin receptors [47,48]. Furthermore, palmitic acid activates JNK via a non-receptor tyrosine kinase Src-dependent pathway [49]. We presume that the intracellular signaling crosstalk, at least in part, accounts for the accelerating effects of palmitic acid and leptin on the innate immune responses of epidermal keratinocytes.

Our findings highlight the essential role of epidermal keratinocytes as responder cells to palmitic acid and leptin. We showed that NHEKs produce pro-inflammatory cytokines, including chemokines, in response to palmitic acid and leptin. Thus, epidermal keratinocytes can be both initiators and amplifiers of the inflammatory milieus by recruiting and responding to immune cells. The importance of epidermal keratinocytes in the pathogenesis of psoriasis has been strikingly illustrated in some murine genetic models. Keratinocyte-specific gene modifications have induced psoriasiform dermatitis—for instance, in K5.Stat3c transgenic mice, *Klk6* transgenic mice, and *K14-Rac1V12^−/+^* mice [20,44,50]. Activation of keratinocytes induces aberrant immune responses in psoriatic skin [51]. Our study showed that the innate immune responses of epidermal keratinocytes could contribute critically to the exacerbated phenotypes of HFD-fed *Apoe*^−/−^ psoriatic mice.

Interestingly, we found that mice with obesity alone or dyslipidemia alone exhibited characteristic expression patterns of pro-inflammatory cytokines, depending on the pathological metabolic status. HFD feeding elevated *Ccl20* expression, whereas *Apoe*-deficiency-mediated dyslipidemia upregulated *Il19* expression in the skin without IMQ application (Appendix A). C-C motif chemokine ligand-20 (CCL20) is a chemokine that recruits C-C chemokine receptor-6-expressing cells, including Th17 and γδT cells [23]. IL-19, a member of the IL-10 cytokine family, amplifies the pro-inflammatory effect of IL-17A on epidermal keratinocytes, including the expression of IL-23p19, β-defensins, and Th17- and neutrophil-attracting chemokines [52]. These findings suggest that metabolic diseases predispose the skin to psoriasis; in such a case, the skin can be considered to be in a pre-psoriatic state. Additionally, the pathogenesis of the pre-psoriatic state may differ depending on the underlying metabolic diseases. In our study, *IL19* gene expression in NHEKs was significantly amplified by palmitic acid (Figure 5a), but not by leptin (data not shown). These findings explain the mechanisms by which *Il19* genes are elevated in the skin of *Apoe*^−/−^ mice, which could support the concept of distinct inflammatory patterns depending on pathological conditions. Collectively, we believe that this concept explains the underlying mechanisms by which the complications of metabolic disorders worsened the pathology of psoriasis in the present study.

Previous studies have reported that HFD significantly enhanced skin thickness in IMQ-induced murine models [13,14,53,54]. Although HFD feeding substantially increased pro-inflammatory cytokine gene expression in the inflamed skin of IMQ-treated mice in our study, HFD did not significantly exacerbate IMQ-induced skin thickness compared to ND feeding (Figure 2b). Differences in the impact of HFD feeding between previous studies and our study might result from differences in experimental settings, such as amount, duration, and location of IMQ application, as well as the composition of HFD. In our experiment, 25 mg/mouse of IMQ was applied unilaterally to the ear for 5 days. In other studies, IMQ at 10 mg [13,53], 62.5 mg [54], or 100 mg [14] per mouse was used and applied on the back [14,54] for 3 days [14] or 6 days [12]. In terms of HFD, the percentage of SFAs was 22.3% in our study. In other reports, percentages of SFAs were higher than that in our study, as can be surmised from the described details [13,14]. Indeed, the severity of IMQ-induced skin changes is affected by the composition of the HFD—especially SFAs and polyunsaturated fatty acids (PUFAs) [14]. In this study, the authors compared the effects of HFD-SFA^high^-PUFA^low^ (crude fat 35.7%) and HFD-SFA^low^-PUFA^high^ (crude fat 21.1%), and showed that HFD-SFA^high^-PUFA^low^ feeding significantly enhanced IMQ-induced epidermal thickness and inflammation, whereas HFD-SFA^low^-PUFA^high^ feeding did not exacerbate skin changes compared to that in standard chow feeding (SFA^low^-PUFA^low^, crude fat 4.5%) [14].

From a clinical perspective, our findings suggest the importance of managing metabolic diseases to prevent psoriatic skin manifestations. Intensive therapeutic interventions for existing metabolic disorders may effectively prevent the development of psoriasis in, for instance, individuals with a family history of psoriasis. Another clinical significance is that elucidating the metabolic-status-dependent pre-psoriatic inflammatory patterns may help in establishing therapeutic strategies tailored to the existing complications for each patient. In addition, some anti-dyslipidemic treatments (e.g., omega-3 supplementation, anti-PCSK9 antibodies, and statins) may have protective effects on the pre-psoriatic inflammation beyond just normalizing aberrant lipid profiles, just like the antidiabetic agent PPARγ inhibitor has an anti-inflammatory capacity [55]. Further animal studies and detailed analyses of clinical data would warrant the establishment of such strategies.

In conclusion, obesity and dyslipidemia synergistically exacerbate psoriatic skin inflammation, and metabolic-disorder-associated inflammatory factors, palmitic acid, and leptin activate epidermal keratinocytes in the presence of inflammatory cytokines. These data clarify the mechanisms underlying the initiation and augmentation of psoriatic inflammation in metabolic diseases. Furthermore, our data emphasize the importance of intensive treatment for accompanying metabolic disorders in patients with psoriasis.

## 4. Materials and Methods

### 4.1. Reagents

IMQ cream (Beselna cream) was kindly provided by Mochida Pharmaceutical (Tokyo, Japan). Recombinant human TNF-α, IL-17A, and leptin were purchased from Chemicon (Temecula, CA, USA), R&D Systems (Minneapolis, MN, USA), and PeproTech (Cranbury, NJ, USA), respectively. The BSA–palmitate saturated fatty acid complex and BSA control were purchased from Cayman Chemical (Ann Arbor, MI, USA). Anti-STAT3 (clone: D1B2J, #30835), anti-phospho-STAT3 (Tyr705, clone: D3A7, #9145), anti-phospho-JNK (clone: 81E11, #4668), anti-total-JNK (#9252), anti-phospho-p38 (clone: D3F9, #4511), anti-total-p38 (clone: D13E1, #8690), anti-phospho-NF-κB p65 (clone: 93H1, #8214), and anti-total-NF-κB p65 (clone: D14E12, #8242) antibodies were purchased from Cell Signaling Technology (Danvers, MA, USA). Anti-IL-23p19 antibody (#ab45420) was obtained from Abcam (Waltham, MA, USA). Anti-actin antibody was purchased from Sigma-Aldrich (A2066, St. Louis, MO, USA).

### 4.2. Mice and Diets

*Apoe*-deficient (*Apoe*^−/−^) mice (C57BL/6J-Apoetm1Unc) were purchased from Charles River Laboratories (Wilmington, MA, USA). Wild-type (WT) C57BL/6J mice were purchased from Charles River Laboratories Japan, Inc. (Yokohama, Japan). All of the mice were housed in groups (3–5 mice per cage) and maintained at 22 °C under 12 h light/12 h dark cycles with free access to water and chow in the animal facility of Kawasaki Medical School (Okayama, Japan).

WT and *Apoe*^−/−^ mice were randomly assigned to ND or HFD groups. The sex ratio of each group was 1:1. Mice assigned to the ND group received standard laboratory food (MF diet; Oriental Yeast Co., Tokyo, Japan), while those assigned to the HFD groups were fed high-fat chow (HFD32; CLEA Japan, Tokyo, Japan). The mice were fed these diets for 10 weeks, from when they were approximately 10 weeks old until the end of the experiment.

### 4.3. IMQ-Induced Psoriasis Model

WT and *Apoe^−/−^* mice fed with ND or HFD were assigned to the IMQ group or Vaseline group, creating eight experimental groups (*n* = 6 each). Mice assigned to the IMQ group were treated with 25 mg of 5% IMQ cream (Beselna cream) on the right ear for five consecutive days. The ear thickness was measured daily using a dial thickness gauge. The mice were dissected one day after the last treatment. The mice were fasted for 6 h before dissection. After euthanasia, blood and skin samples were collected (Figure 1a).

Equal numbers of males and females were used in each group. Since both males and females responded to the experimental manipulations similarly, all of the data except for the body weights were combined and analyzed for males and females.

All animal experiments were approved by the Institutional Safety Committee for Recombinant DNA Experiments (19–31, 19–32) and the Institutional Animal Care and Use Committee of Kawasaki Medical School (20-073). All experimental procedures were conducted according to the institutional and NIH guidelines for the humane use of animals.

### 4.4. Histological Analyses of the Ears

The ear tissues were fixed in 4% paraformaldehyde for 2 days and then embedded in paraffin. Sections from the skin samples were stained with hematoxylin and eosin using standard methods. The epidermal thickness was quantified using the representative images by averaging the line measurements of the epidermis, excluding the stratum corneum, using ImageJ (NIH, Bethesda, MD, USA), as described previously [56]. Ten representative areas of the epidermis were measured and their average value was calculated.

### 4.5. Immunohistochemistry

STAT3, p-STAT3, and IL-23p19 expression was determined in skin tissues by immunohistochemical staining. Tissue sections were deparaffinized and rehydrated. Epitope retrieval was performed by heating the slides three times for 5 min at 100 °C in 0.01 M citrate buffer (pH 6). Endogenous peroxidase activity was inhibited using 3.0% hydrogen peroxide in methanol for 10 min. After blocking with normal goat serum, the slides were incubated with anti-STAT3, p-STAT3, or IL-23p19 primary antibodies (1:1000 dilution) overnight at 4 °C, followed by 60 min of incubation with a horseradish peroxidase-conjugated goat anti-rabbit secondary antibody. Finally, the indicated protein expression was visualized using a 3,3-diaminobenzidine substrate–chromogen system (Nichirei Biosciences Inc., Tokyo, Japan). Tissue sections stained with isotype controls were used as the negative controls.

### 4.6. Measurement of Biochemical Parameters

The blood samples were centrifuged to separate the serum, and serum concentrations of total cholesterol and triglycerides were measured using a biochemical autoanalyzer (DRICHEM 7000V, FUJIFILM Medical Co., Ltd., Tokyo, Japan). Blood glucose levels were measured using a glucometer (ForaCare Japan Co., Ltd., Tokyo, Japan). Free fatty acid levels were measured by LabAssay NEFA (FUJIFILM Wako Pure Chemical Co., Ltd., Tokyo, Japan).

### 4.7. Multiplex Analysis for Adipokines

Adipokine profiles in mouse sera were determined using a Milliplex mouse adipokine magnetic bead panel (Merck Millipore, Billerica, MA, USA), as reported previously [57]. The measured adipokines were leptin, resistin, IL-6, monocyte chemotactic protein-1, and TNF-α.

### 4.8. ELISA for IL-17A

IL-17A concentrations in mouse sera were measured using a murine IL-17A ELISA MAX™ Deluxe kit (BioLegend; San Diego, CA, USA), following the manufacturer’s protocol. Optical density at 450 nm was measured using a microplate reader (Varioskan Flash), and the IL-17A concentration in each sample was calculated based on a standard curve.

### 4.9. Real-Time Quantitative Polymerase Chain Reaction

Real-time qPCR was performed as previously described [58,59]. Total RNA was extracted from the right ear or cultured cells using RNAiso Plus (Takara Bio, Shiga, Japan), and then solubilized in ribonuclease-free water. Complementary DNA (cDNA) was synthesized using the PrimeScript RT Reagent Kit (Takara Bio). qPCR reactions were performed using TB Green PCR Master Mix (Takara Bio) with the StepOnePlus Real-Time PCR System (Thermo Fisher Scientific, Waltham, MA, USA) or the QuantStudio1 System (Thermo Fisher Scientific). Gene expression levels relative to *Gapdh* for murine samples and *GAPDH* for human-derived cells were calculated by the ∆∆Ct method and normalized to control samples. We also tested *Hprt* and *HPRT* as reference genes for murine and human samples, respectively, and confirmed that the reference genes yielded similar results to those analyzed by *Gapdh* or *GAPDH*. qPCR analysis was performed using the primers listed in Appendix A. All qPCR reactions yielded products with single-peak dissociation curves.

### 4.10. Cell Culture and Stimuli

NHEKs were obtained from Thermo Fisher Scientific (C-001-5C) and grown in serum-free EpiLife Medium (Thermo Fisher Scientific) containing 0.06 mM Ca^2+^ and EpiLife Defined Growth Supplement (Thermo Fisher Scientific) at 37 °C/5% CO_2_. Cultures were maintained for up to eight passages in this medium with the addition of 100 IU/mL penicillin, 100 µg/mL streptomycin, and 0.25 µg/mL amphotericin B. NHEKs were grown in 12-well flat-bottomed plates. Upon reaching 70% confluence, NHEKs were stimulated with TNF-α (30 ng/mL), IL-17A (30 ng/mL), leptin (30 ng/mL), BSA–palmitate saturated fatty acid complexes (100 µM), and/or BSA controls for fatty acid complexes for up to 24 h.

### 4.11. Western Blotting

NHEKs were washed with ice-cold phosphate-buffered saline and lysed with a radioimmunoprecipitation assay lysis buffer (Sigma-Aldrich) containing protease and phosphatase inhibitor cocktails (Sigma-Aldrich). Protein concentrations were determined using a bicinchoninic acid protein assay kit (Thermo Fisher Scientific). Protein samples (8 μg/lane) were resolved by SDS–PAGE and transferred to polyvinylidene fluoride membranes. After blocking with 5% BSA in Tris-buffered saline with Tween 20, the membranes were incubated with primary antibodies, followed by further incubation with the appropriate horseradish peroxidase-conjugated species-specific secondary antibodies. All primary antibodies were diluted at a ratio of 1:1000, and secondary antibodies were diluted at 1:10,000. Bands were detected using a SuperSignal West chemiluminescent substrate (Thermo Fisher Scientific) and visualized using ImageQuant LAS-4000 (GE Healthcare, Little Chalfont, UK). Actin was used as a loading control to normalize the amount of protein. The intensities of the bands were quantified using ImageJ.

### 4.12. Statistical Analysis

All values are presented as the mean ± standard deviation, except for the values of the time course of the ear thickness variation, which are presented as the mean ± standard error of the mean. One-way analysis of variance followed by Tukey’s post hoc test was used to compare three or more groups. Dunnett’s post hoc test was used for the data analysis shown in Figure 6a. All statistical analyses were performed using GraphPad Prism 4 (GraphPad Software, San Diego, CA, USA). Statistical significance was set at *p* < 0.05.

## Figures and Tables

**Figure 1 ijms-23-04312-f001:**
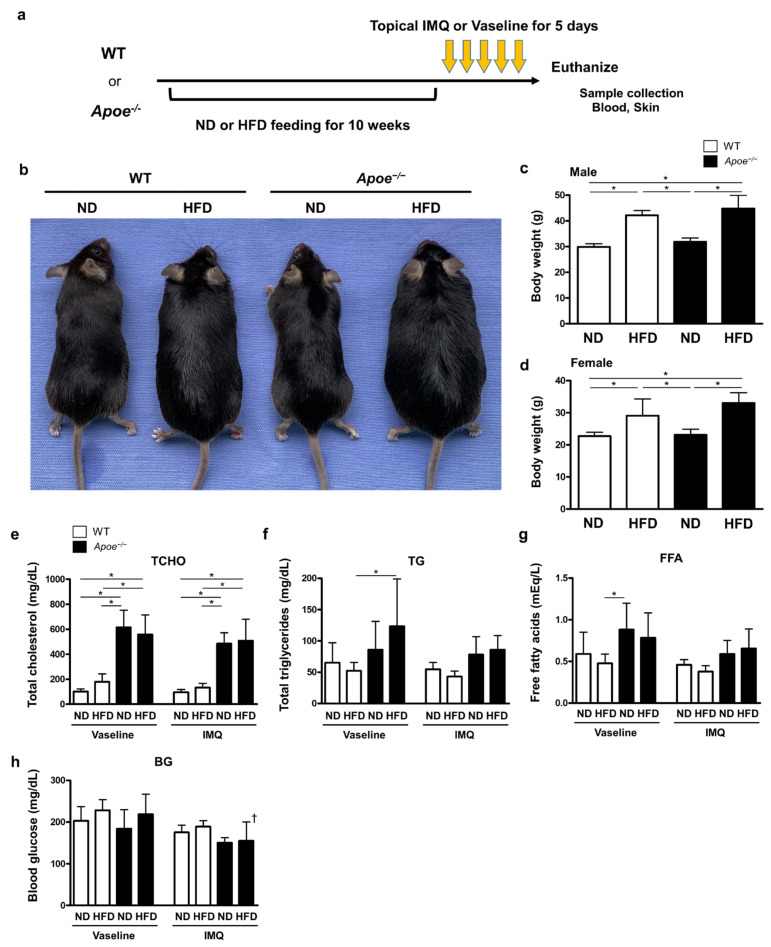
High-fat diet feeding and *Apoe* deficiency provoked remarkable metabolic changes. (**a**) Study design: Ten-week-old WT and *Apoe*^−/−^ mice were fed with ND or HFD for 10 weeks. IMQ cream (25 mg/mouse) or Vaseline (as control) were topically applied on the ear for 5 consecutive days (*n* = 6 per group). One day after the last treatment, fasting sera and ear tissues were collected. (**b**) Mouse appearance. (**c**,**d**) Body weights of male and female mice. (**e**–**h**) Serum metabolic parameters: Serum concentrations of total cholesterol, triglyceride, free fatty acid, and blood glucose were determined by biochemical assays. Values are presented as the mean ± standard deviation; * *p* < 0.05, ^†^
*p* < 0.05, compared to Vaseline-treated mice. IMQ, imiquimod; WT, wild-type; ND, normal diet; HFD, high-fat diet; TCHO, total cholesterol; TG, triglyceride; FFA, free fatty acid; BG, blood glucose.

**Figure 2 ijms-23-04312-f002:**
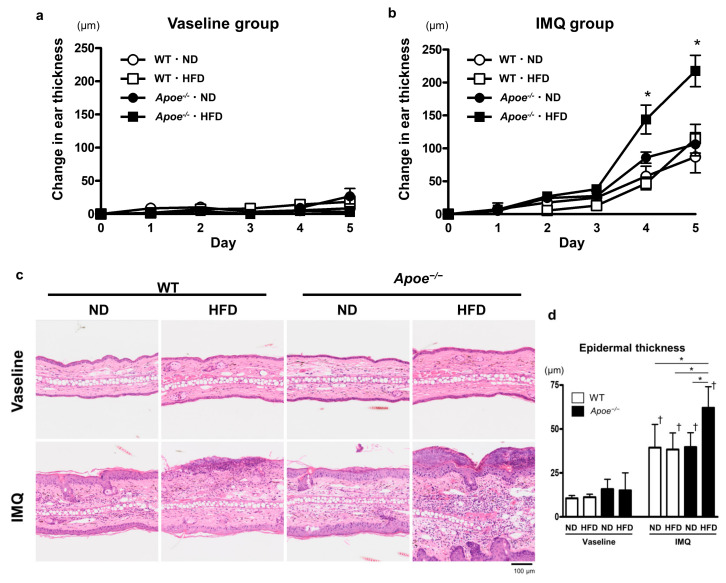
Synergistic effects of obesity and dyslipidemia on psoriatic skin inflammation: (**a**,**b**) Changes in ear thickness. Wild-type and *Apoe*^−/−^ mice were fed with a normal diet or a high-fat diet and treated with imiquimod cream (25 mg/mouse) or Vaseline (control) (*n* = 6 per group). Values are shown as the mean ± standard error of the mean; * *p* < 0.05, compared to other groups at indicated time points. (**c**) Representative images of hematoxylin- and eosin-stained ears of indicated mice. Bar = 100 μm. (**d**) Histological analysis of epidermal thickness. Values are presented as the mean ± standard deviation; * *p* < 0.05, ^†^
*p* < 0.05, compared to Vaseline-treated mice. IMQ, imiquimod; WT, wild-type; ND, normal diet; HFD, high-fat diet.

**Figure 3 ijms-23-04312-f003:**
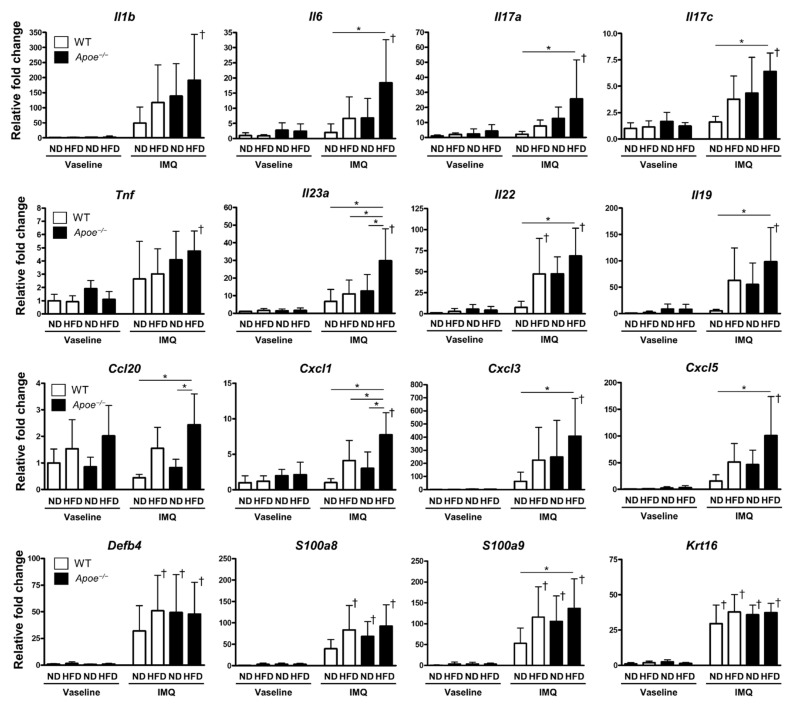
Alterations in the mRNA expression of cytokines, chemokines, and antimicrobial peptides associated with psoriasis. RNA samples were extracted from the ear tissues of wild-type and *Apoe*^−/−^ mice fed with a normal diet or a high-fat diet and treated with imiquimod cream (25 mg/mouse) or Vaseline (control) (*n* = 6 per group). mRNA expression levels were determined by quantitative PCR (qPCR). The levels were calculated relative to *Gapdh* and normalized to the expression level of Vaseline-treated wild-type mice fed with a normal diet. Values are presented as the mean ± standard deviation; * *p* < 0.05, ^†^
*p* < 0.05, compared to Vaseline-treated mice. IMQ, imiquimod; WT, wild-type; ND, normal diet; HFD, high-fat diet.

**Figure 4 ijms-23-04312-f004:**
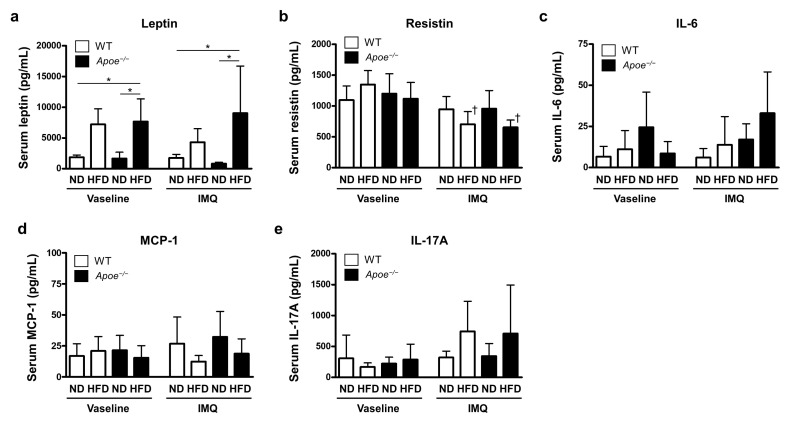
Measurement of multiple adipokine levels in the mouse serum: (**a**–**d**) Serum levels of leptin, resistin, IL-6, and MCP-1. Serum samples were collected from wild-type and *Apoe*^−/−^ mice fed with a normal diet or a high-fat diet and treated with imiquimod cream (25 mg/mouse) or Vaseline (control) (*n* = 6 per group). Serum adipokine levels were measured by multiplex assay. (**e**) Serum IL-17A concentrations determined by ELISA. Values are presented as the mean ± standard deviation; * *p* < 0.05, ^†^
*p* < 0.05, compared to Vaseline-treated mice. IMQ, imiquimod; WT, wild-type; ND, normal diet; HFD, high-fat diet.

**Figure 5 ijms-23-04312-f005:**
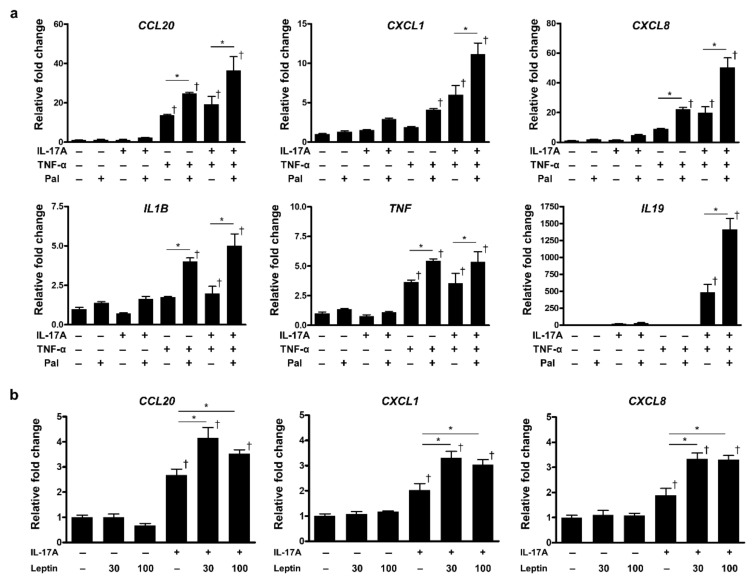
Effects of palmitic acid and leptin on epidermal keratinocytes: (**a**) mRNA expression after palmitic acid treatment. Normal human epidermal keratinocytes (NHEKs) were stimulated with IL-17A (30 ng/mL), TNF-α (30 ng/mL), and palmitic acid (Pal; 100 µM) for 4 h. *IL19* expression was determined after 24 h. (**b**) mRNA expression after leptin treatment. NHEKs were incubated with IL-17A (30 ng/mL) with/without leptin (30 or 100 ng/mL) for 24 h. Gene expression levels relative to *GAPDH* were calculated and normalized to those of non-stimulated controls. Values are presented as the mean ± standard deviation; * *p* < 0.05, ^†^
*p* < 0.05, compared to non-stimulated controls.

**Figure 6 ijms-23-04312-f006:**
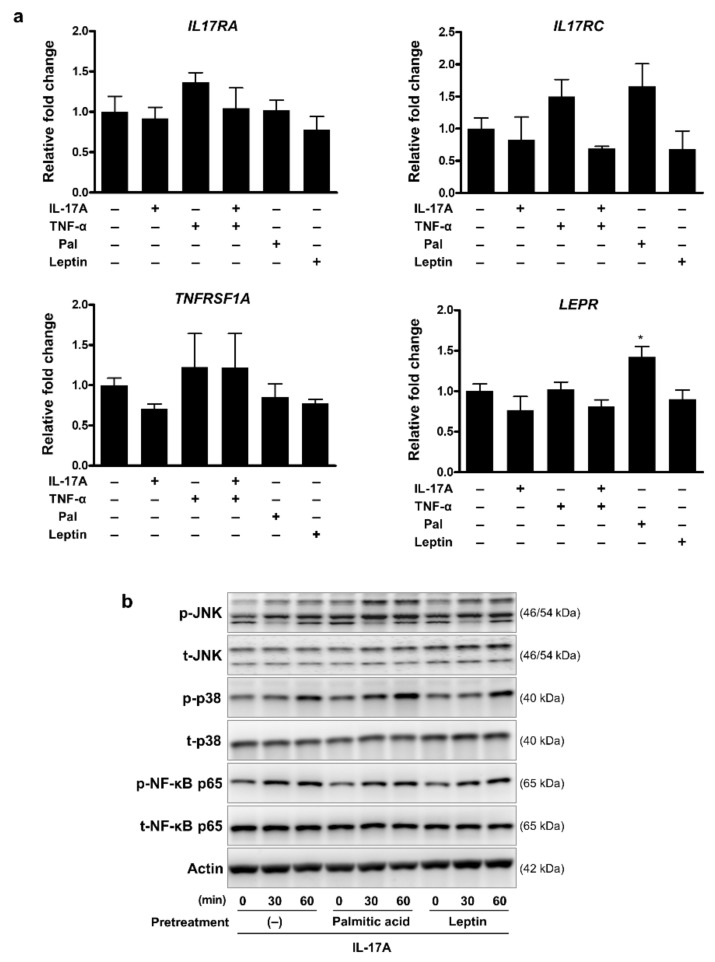
Effects of palmitic acid and leptin on receptors and signaling pathways in epidermal keratinocytes: (**a**) mRNA expression of receptors. NHEKs were treated with the indicated cytokines and palmitic acid. The mRNA expression levels of each receptor were determined 24 h after the indicated stimulation. Values are presented as the mean ± standard deviation; * *p* < 0.05, compared to non-stimulated cells. (**b**) Representative images of Western blot analysis. NHEKs were pretreated with palmitic acid (100 µM) or leptin (100 ng/mL) for 2 h, and then treated with IL-17A (30 ng/mL) for the indicated time. Phospho- or total-JNK, p38, and NF-κB p65 were detected using specific antibodies. Actin was used as a loading control.

## Data Availability

The data that support the findings of this study are available from the corresponding author upon reasonable request.

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
