# Peer review of "Obesity and Dyslipidemia Synergistically Exacerbate Psoriatic Skin Inflammation"

_ijms, 2022, doi:10.3390/ijms23084312_

Round 1

Reviewer 1 Report

The article is very interesting with a well defined structure and a fully detailed background too.  Informations about the previous study findings are clearly presented and described.

Major issue:

-As reported by the authors “In terms of HFD, the percentage of saturated fatty acids was 22.3% In other reports, percentages of saturated fatty acids were higher than that in their study” Did the authors consider to evaluate different percentage of fatty acids in order to have a better understanding of their role?

-Can the author be more accurate on the severity of IMQ-induced skin changes? this is “affected by the composite of HFD, especially  saturated fatty acids and polyunsaturated fatty acids” as correctly said in the paper but- how? Maybe some more references can be helpful.

-Authors didn’t observe significant changes in blood glucose levels across the studied groups

Did the authors consider to evaluate these differences depending on feeding mouse time?

-As in the paper “Patients with psoriasis often have multiple comorbidities. Additionally, the number of metabolic factors have been shown to correlate positively with the risk of psoriasis. Although individual metabolic conditions potentially exacerbate psoriatic skin changes in murine models, the synergistic effects of metabolic conditions have not been investigated”

Did the author consider to deepen this important issue?

Minor Issue:

-“Further animal studies and detailed analyses of clinical data would warrant the  establishment of such strategies” Did the authors begin to evaluate this model and/or can indicate some recent studies about this?

Author Response

Reviewer#1

The article is very interesting with a well defined structure and a fully detailed background too.  Informations about the previous study findings are clearly presented and described.

Response: We appreciate the reviewer’s positive comments.

Major issue:

-As reported by the authors “In terms of HFD, the percentage of saturated fatty acids was 22.3% In other reports, percentages of saturated fatty acids were higher than that in their study” Did the authors consider to evaluate different percentage of fatty acids in order to have a better understanding of their role?

Response: We thank the reviewer’s suggestion. I am afraid we have not tested the different percentages of saturated fatty acids by ourselves. As the reviewer suggested, such experiments would reveal the essential roles of saturated fatty acid in the pathogenesis of psoriasis. We would like to plan the experiments for future research.

-Can the author be more accurate on the severity of IMQ-induced skin changes? this is “affected by the composite of HFD, especially  saturated fatty acids and polyunsaturated fatty acids” as correctly said in the paper but- how? Maybe some more references can be helpful.

Response: As the reviewer suggested, we have described the detailed results which were reported in the previous study (Ref #14 Journal of Investigative Dermatology 2018, 138, 1999-2009.). The description was added to Discussion, lines 357-361. Also, as the reviewer pointed out, we have added the description regarding the possible mechanisms by which palmitic acid exacerbates psoriasis. The description was added to Discussion, lines 295-303.

-Authors didn’t observe significant changes in blood glucose levels across the studied groups

Did the authors consider to evaluate these differences depending on feeding mouse time?

Response: As the reviewer pointed out, high-fat diet feeding could affect insulin resistance and blood glucose levels. Although the fasting blood glucose levels were not significantly different across the studied groups, some differences in blood glucose levels might be observed at different timings, for example, shortly after the feeding. A glucose tolerance test might be useful to detect the potential differences in the insulin resistance of the mice. This is the future direction to be investigated.

-As in the paper “Patients with psoriasis often have multiple comorbidities. Additionally, the number of metabolic factors have been shown to correlate positively with the risk of psoriasis. Although individual metabolic conditions potentially exacerbate psoriatic skin changes in murine models, the synergistic effects of metabolic conditions have not been investigated”

Did the author consider to deepen this important issue?

Response: We thank the reviewer for this question. In our study, we have tested hyperlipidemic mice (Apoe-/- mice) and obese mice (HFD feeding) and showed the synergistic effect of hyperlipidemia and obesity in the exacerbation of psoriatic skin changes. To deepen the synergistic effects of metabolic disorders, we need to consider several other metabolic disorders, such as diabetes mellitus, hypertension, and hyperuricemia, as well as obesity and dyslipidemia. We are going to evaluate the synergistic effects of some of the metabolic disorders in the next project.

Minor Issue:

-“Further animal studies and detailed analyses of clinical data would warrant the  establishment of such strategies” Did the authors begin to evaluate this model and/or can indicate some recent studies about this?

Response: We thank the reviewer for this question. We have not started the detailed analyses. Now, we plan to perform RNAseq analysis to compare gene expression in the skin between hyperlipidemic mice (Apoe-/- mice) and obese mice (HFD feeding). We hope the analyses help to clarify specific pre-psoriatic inflammatory patterns in the skin. After that, we would like to proceed with the gene expression analysis in human skin samples, which would reveal more concrete findings.

Reviewer 2 Report

The authors submitted a research article with the aim of elucidating pathogenic effects of metabolic disease-associated inflammatory factors on the innate immune responses of epidermal keratinocytes. They used Apoe-deficient hyperlipidemic mouse model and administered a high-fat diet for 10 weeks to induce obecity. The authors found that  the treatment of  neonatal human epidermal keratinoc with palmitic acid and leptin amplified pro-inflammatory responses in combination with TNF-α and IL-17A and that pretreatment with palmitic acid and leptin enhanced IL-17A-mediated c-Jun N-terminal kinase phosphorylation. These findings appear to be intriguing and practically useful. However, I would like to pot forward several question to discuss.

  1. The term "hyperlipidemia" seems to be pathogenetic, but not widely used in routine practice. In addition, the authors, in fact, described an effect of dyslidemia on epidermal keratinocytes. Please, consider about turning the term hyperlipidemia into dyslipidemia.
  2. Please, add statistics data regarding an coincidance of psoriasis to dyslipidemia (line 47-48).
  3. The authors resumed that the data have the importance of managing metabolic diseases to prevent psoriatic skin manifestation, but they did not indicate in which ways the therapy should be performed. Please, give this proposal in detail (omega-3 supplementation, anti-PCSK9, statins, etc)

Author Response

Reviewer#2

The authors submitted a research article with the aim of elucidating pathogenic effects of metabolic disease-associated inflammatory factors on the innate immune responses of epidermal keratinocytes. They used Apoe-deficient hyperlipidemic mouse model and administered a high-fat diet for 10 weeks to induce obecity. The authors found that  the treatment of  neonatal human epidermal keratinoc with palmitic acid and leptin amplified pro-inflammatory responses in combination with TNF-α and IL-17A and that pretreatment with palmitic acid and leptin enhanced IL-17A-mediated c-Jun N-terminal kinase phosphorylation. These findings appear to be intriguing and practically useful. However, I would like to pot forward several question to discuss.

Response: We appreciate the reviewer’s positive comments.

  1. The term "hyperlipidemia" seems to be pathogenetic, but not widely used in routine practice. In addition, the authors, in fact, described an effect of dyslidemia on epidermal keratinocytes. Please, consider about turning the term hyperlipidemia into dyslipidemia.

Response: We thank the reviewer’s suggestion. As the reviewer pointed out, the term dyslipidemia is widely used in clinical practice to describe aberrant lipid profiles. We have turned the term hyperlipidemia into dyslipidemia.

  1. Please, add statistics data regarding an coincidance of psoriasis to dyslipidemia (line 47-48).

Response: As the reviewer suggested, we appreciate the reviewer’s suggestion. We have added the description by citing additional literatures in Introduction, lines 47-51.

  1. The authors resumed that the data have the importance of managing metabolic diseases to prevent psoriatic skin manifestation, but they did not indicate in which ways the therapy should be performed. Please, give this proposal in detail (omega-3 supplementation, anti-PCSK9, statins, etc)

Response: We thank the reviewer for this suggestion. We have shown that systemic metabolic abnormalities could affect the severity of psoriasis and that intracellular signaling crosstalk is, at least in part, involved in the mechanisms. Because we have not focused on drugs in this study, it would be difficult to propose specific drugs for better treatments targeting metabolic disorders associated with psoriasis. We assume that any drugs that improve metabolic status would be beneficial for improving psoriasis. As of now, it is not clear whether some specific drugs have an additive therapeutic effect via modulation of the proinflammatory pathways. This point should be clarified in future studies because it has significant clinical implications, as the reviewer pointed out. We have added the description in Discussion, lines 368-371.